# Perceptions and Experiences of Adolescents with Mental Disorders and Their Parents about Psychotropic Medications in Turkey: A Qualitative Study

**DOI:** 10.3390/ijerph19159589

**Published:** 2022-08-04

**Authors:** Gül Dikec, Cansın Kardelen, Laura Pilz González, Marjan Mohammadzadeh, Öznur Bilaç, Christiane Stock

**Affiliations:** 1Institute of Health and Nursing Sciences, Charité—Universitätsmedizin Berlin, Humboldt Universität zu Berlin and Freie Universität, 13353 Berlin, Germany; 2Department of Nursing, Faculty of Health Sciences, Fenerbahce University, Istanbul 34758, Turkey; 3Department of Child and Adolescent Psychiatry, Hafsa Sultan Medical School Hospital, Celal Bayar University, Manisa 45030, Turkey; 4Unit for Health Promotion Research, University of Southern Denmark, 6705 Esbjerg, Denmark

**Keywords:** adherence, adolescent psychiatry, psychotropic drugs

## Abstract

This descriptive phenomenological study aimed to evaluate the perception and experiences of adolescents with mental disorders and their parents about the use of and adherence to psychotropic medications. A total of 12 semi-structured interviews with adolescents between the ages of 12 to 18 who were attending an outpatient psychiatric clinic for children and adolescents and 12 interviews with parents were conducted between October 2021 and January 2022 in Manisa, Turkey. Colaizzi’s phenomenological interpretation method was used for the analysis of the participants’ statements. Our study highlights the main positive effects of psychotropic medication and barriers to medication intake and adherence. Positive effects included symptom management and health improvement. Barriers varied from those directly linked to medication effects (e.g., negative side effects or lack of perceived effect) to personal barriers (e.g., forgetting to take medication or feelings of not being oneself due to medication intake) and societal barriers. In general, the barriers were reflected in concerns related to long-term consequences, such as medication dependence, and in concerns about diminished life prospects. Possible recommendations to improve the use of and adherence to psychotropic medication among adolescents include educating adolescents and parents not only about treatment options but also about mental disorders.

## 1. Introduction

Adherence to medication describes the extent to which individuals take their medication regimen as prescribed by their health care provider [1]. While psychotropic medication is an essential component of the treatment of mental disorders in the early stages [2,3], adherence to it has often proven to be challenging [4,5]. Nonadherence to psychotropic medication has shown to increase relapses, psychiatric rehospitalisation, and morbidity not only among adults but also among adolescents [5,6]. Particularly for adolescents with mental disorders, adherence to psychotropic medication is relatively low with only about two-thirds adhering to the prescribed medication [2,6,7,8]. Such limited adherence rate in this age group is quite worrisome, as nonadherence among adolescents can have specific disruptive effects on their life and development, such as social isolation, school difficulties, criminal records, unintended pregnancies, sexually transmitted diseases, and substance abuse, among others [2,3].

Nonadherence may be associated with a number of factors, such as sociodemographic and sociocultural aspects (e.g., stigmatisation, social and environmental support) [6,8]. In addition, clinical factors (e.g., diagnosis and severity of mental disorder, comorbidities), medication-related factors (e.g., tolerability of medication, side effects), and factors related to access to health care (e.g., availability of mental health services) and interaction with health professionals (e.g., communication with mental health professionals) also play a relevant role [2,3,9,10,11]. A systematic review on the topic has shown that adherence to medication is generally related to the positive attitudes of patients and their families towards treatment, psychotherapy, and their level of information [8]. As for nonadherence, it has been shown to be strongly related to the severity of the diagnosed mental disorder(s). Especially disorders such as attention deficit hyperactivity disorder (ADHD) or substance abuse show higher levels of nonadherence to medication [6].

While the views, perceptions, and experiences of adults with mental disorders on the use of psychotropic medication have been thoroughly explored in the literature, little attention has been paid to those of adolescents [7,8,9,12,13]. It is well known that the experiences of adults with mental disorders cannot be generalised to younger populations, as the experiences of adolescents may be influenced by other factors. In addition to the ‘typical’ problems of being an adolescent (e.g., identity formation, body and developmental issues, peer pressure [8]), adolescents with mental disorders also have to deal with the challenges of having such a condition and balancing different forms of treatment. Furthermore, as minors, their right and capacity for autonomous decision making is limited to that of their parents or legal guardians. In this sense, their parents or healthcare professionals can decide to what extent they are included in the decision-making process [8,14]. This could have consequences, such as being forced to take medication without their consent or, on the contrary, not having the support of their parents or guardians to do so [4,8,15].

One of the few systematic reviews that focus on qualitative evidence on medication adherence among adolescents with mental disorders is a study by McMillan et al. [8]. This study highlights both the negative and positive experiences of adolescents in taking and adhering to psychotropic medication. Factors such as lack of autonomy and confrontation with their own medication regimen and thus with the benefits and (side) effects of medication, as well as the responsibility and burden it requires, were highlighted as key issues influencing the adherence or nonadherence to medication [8]. In addition, family factors were shown to play a significant role in influencing the experiences of taking and adhering to psychotropic medication, as decisions about the adolescent’s health and well-being depend on them. Additional studies have also shown the strong relation between adherence rate of medication and family influences [3,9,10]. Timlin et al. highlighted the positive influence of a well-functioning relational family or social networks on adherence in adolescents [9], which was also emphasised by Häge et al. [10], who pointed out the negative relation between medication nonadherence and parental perceptions. The strong association between medication adherence and family dynamics and influence reflect that it is necessary to determine not only the perceptions, attitudes, and experiences of adolescents with mental disorders about psychotropic medication but also those of their parents.

Considering that sociocultural aspects and religion are also important components in relation to individuals’ perceptions of health and illness, the concept of family and its influence on adolescents becomes even more important [16]. This is especially true in the sociocultural context in which this study was conducted (i.e., Turkey), where family plays an important role in the lives of adolescents. However, the few studies reporting on the topic have been mostly conducted in Western countries, where not only these social dynamics vary but also cultural and religious ones. Therefore, in order to develop effective and useful techniques to improve adherence to psychotropic medication and reduce health expenditure, it is necessary to understand what influences medication adherence or nonadherence among adolescents with mental disorders and their families, in this case, their parents [6]. Doing so allows us to take into account the sociocultural background and context.

The results of this study may inform the development of psychosocial interventions to enhance the medication adherence of adolescents based on the context of Manisa in Western Turkey, and thus provide insights on this already underreported topic, especially in non-Western contexts. Based on this background, the present study aimed to evaluate the perception and experiences of adolescents with mental disorders and their parents about the use of and adherence to psychotropic medications.

## 2. Materials and Methods

### 2.1. Study Design

A descriptive phenomenological design was used for the study as it provides an appropriate approach to the study of events, experiences, perceptions, and orientations of individuals [17,18]. In addition, the Consolidated Criteria for Reporting Qualitative Studies (COREQ) was used in the reporting of this study as a tool commonly used in reporting qualitative studies [19] (Appendix A).

### 2.2. Participants

The sample of the study included adolescents between the ages 12 to 18 who were attending an outpatient psychiatric day clinic for children and adolescents in Manisa in Western Turkey. In addition, at least one of their parents was included as study participant. The inclusion criteria for the participation of the adolescents included the diagnosis of any mental health disorder according to the *Diagnostic and Statistical Manual of Mental Disorders, Fifth Edition* (DSM-V) and the use of psychotropic medications for at least 3 months. In addition, participants had to give their consent to take part in the study. Parents between the ages of 18 to 65 and not diagnosed with a mental disorder according to the DSM-V were eligible to participate. Adolescents with acute symptoms (e.g., hallucinations and delusions) or who had an episode of mental disorder in an inpatient child and adolescent psychiatric day clinic during the study period and those who had been taking medication for less than 3 months were excluded from the study.

A purposive sampling method with a small sample size of 5 to 25 participants was carried out based on the recommendation of previous studies for the suitability of phenomenological studies [20]. Data were collected to the point of data saturation, that is, to the point in the research process where no new information was discovered in the data analysis [20].

### 2.3. Data Collection

In the first step of the data collection procedures, a semi-structured interview guide was developed by the research team according to the literature on this topic [4,5]. The interview guide contained 19 questions divided into four main categories: (1) sociodemographic information, (2) psychotropic medications, (3) history of mental disorders of adolescents, and (4) perceptions, opinions, and experiences with the use of psychotropic medications for the mental disorder. In the third category, the participants were asked directly about substance and alcohol use. No measurement scale was used to collect substance use information. Based on two studies by Brown et al. [4,5] describing the patient and medical-condition-related factors that affect the adherence among young people, guiding questions in this category were formulated. For this, open-ended questions, such as ‘What do you think about this medication you are using/your child is using?’ and ‘How do you think the use of these drugs affects you/your child?’ were asked to understand the feelings, perceptions, and experiences of adolescents and their parents about psychotropic medication. After developing the interview guide, pilot interviews were conducted with two adolescents and their parents. These were not included in the data of our study. No changes to the interview guide were made after the pilot interviews.

The data collection took place in an outpatient psychiatric day clinic for children and adolescents between October 2021 and January 2022. Adolescents and their parents were informed about the study and provided written consent to participate in the study. A total of three sessions were carried out with the participants. In the first session, a senior female child and adolescent psychiatrist (Ö.B.) (medical Doctor (MD)) informed the participants about the study and asked to sign the information and consent forms. As the interviews were audio-recorded, the consent of the adolescents and parents to the recording was also obtained before the interview. Then, eligible participants were referred to a junior female child and adolescent MD (C.K.), who carried out the interviews. The participants were not familiar with the interviewer before. A team member not known to the interviewees who had knowledge about qualitative studies was selected so that adolescents and their families could openly express their feelings and thoughts. The interviewer is trained in therapeutic interviews with children and adolescents with mental disorders and their families in a university hospital in Turkey and conducted all in-depth interviews without another person being present. The interviews were held in a separate room according to the availability of the hospital or clinic. The individual and in-depth interviews were performed face-to-face without any third parties. All interviews were carried out by the same person (C.K.). The interviews lasted 30 min on average and were conducted in Turkish. During the third session, the participants were asked to read and make any necessary corrections to the transcript in order to improve the validity of the data. When the data started to become repetitive, in other words, when the data reached saturation, no more interviews were conducted. Field notes were not made during the interviews.

### 2.4. Data Analysis

In a phenomenological approach, data analysis aims at revealing the experiences and meanings of participants [17]. In this study, Colaizzi’s [21] phenomenological interpretation method was used for the evaluation of the statements of the participants. Thus, we followed Colaizzi’s steps to phenomenological data analysis and implemented them as follows: (1) The interviews were transcribed in their original Turkish language and read in their full length. In addition, each participant read the transcript of their interview to ensure its validity. (2) The transcripts were read multiple times to understand the meanings of the emotions and experiences conveyed in the interviews, and thus extract first significant statements. (3) The statements that directly related to the phenomenon of interest were coded in the qualitative software MAXQDA and (4) were grouped into subthemes and themes by both research studies. Finally, (5) two researchers (G.D., Ö.B.) translated the subthemes and expressions to English independently and checked for inconsistencies. After the translation, we could not return to the participants to validate the themes and subthemes. For this reason, we skipped this step in the analysis, but we expect that by checking the transcription of the interviews in the original language by the participants themselves, we can still be accountable to them. The backgrounds of the two researchers who carried out the analysis are in psychiatric and mental health nursing (G.D.) and in child and adolescent psychiatry (Ö.B.), in Turkey, which allowed for an in-depth understanding and translation of the collected data. For the sociodemographic data, a descriptive analysis was carried out in SPSS (26.00).

### 2.5. Ethical Consideration

Ethical approval was obtained from the Ethical Committee of Fenerbahce University (protocol code 2021-6 and 6 October 2021 date of approval). In addition, an institutional permission was obtained from the Celal Bayar University head physician. Written consent was obtained from the adolescents and their legal guardians as well as from the parents for participating in the study. Additionally, the data were obtained from the adolescents in compliance with the Personal Data Protection Law (2016). The names and genders of the participants were removed from the quotes to protect the participants’ rights and ensure their anonymity.

## 3. Results

A total of 24 participants, 12 adolescents and 12 parents, participated in this study. One adolescent and 1 parent dropped out of the study. While they initially accepted to participate in this study in the first session, they did not attend the second one. Additionally, 1 parent was interviewed, and her/his child refused to participate. This interview was not included in this study. From the analysis of the interviews carried out, 472 initial codes were derived, which were grouped into four main themes, which included aspects on the benefits and disadvantages of treatment, barriers to treatment adherence, and the future perspectives, worries, and hopes of both the adolescents and parents. In general, only few of the subthemes differed between the adolescents and parents.

### 3.1. Participants’ Characteristics

#### 3.1.1. Adolescents

The majority of the participating adolescents were female (*n* = 9), their average age was 14.17 (±1.46) years, and all of them attended school. Most of the adolescents perceived their economic status as middle class (66%) and lived with their families (Table 1). More than half of the sample used antidepressants (58.3%), and the two most frequent diagnoses were depressive (33.4%) and anxiety (33.4%) disorders. Around one-quarter of the adolescents had a physical illness, which were mainly were respiratory and urinary diseases. In addition, about half of the participants reported having attempted suicide (Table 2).

#### 3.1.2. Parents

The average age of the parents was 42.42 (±6.09) years. Most of them were female and the mothers of the adolescents (83.3%), nearly all of them were married (91.7%), half of them were employed (50%), and the majority of them perceived their economic status as middle class (83.4%). Nearly half of the parents had a physical disease, with hypertension and diabetes among the most frequent ones (Table 1).

### 3.2. Thematic Analysis

The analysis identified the same four themes for both the adolescents and their parents, which are, therefore, presented together. In the first theme, ‘benefits of the treatment’, both the adolescents and their parents expressed the advantages of the treatment. After sorting out the benefits and the advantages of the treatment, the adolescents and their parents stated possible disadvantages of the medication use in the second theme, ‘disadvantages of the treatment’. The obstacles and challenges during and caused by the treatment were often mentioned and form the third main theme, ‘barriers to medication adherence’. Finally, under the theme of ‘future dreams’, the participants of both groups expressed their future perspectives, worries, and wishes regarding the use of medications (Table 3).

#### 3.2.1. Theme 1: Benefits of Treatment

Regarding the first theme, some of the adolescents and parents mentioned that taking medicine was beneficial for improving their or their children’s health condition and, as a result, improved the quality of their lives. They specifically described the differences between their health conditions before and after taking the medications. They also highlighted the positive effects of controlling their symptoms, such as anxiety.

*In the past, I could not go out of the house. Even when I went to the market, I would have a stomach ache, nausea, diarrhoea, and a very severe headache. I could not stay in the classroom. I could not even attend a virtual class. I did not want my name to appear there, and I was so anxious. The medicine has been very good for me*.(Adolescent 1)

Accordingly, both participant groups explained the positive influence of symptom reduction on their mental health and improvement in their quality of life, especially their social life. The adolescents explained that shifting the focus from symptoms to other aspects of their daily life increased their self-confidence and effectiveness in coping with life’s challenges. In line with this, adolescents with ADHD mentioned performing more effectively and being more focused on their school lessons than before.

The parents also described the positive effects of their children’s medication intake in reducing their symptoms, as they specifically pointed out the problems they caused them in their family’s daily life. These included the challenges of caring for their children at home and the negative effects of the situation on the relationship between them as parents and their children. They considered the improvement of the quality of their family life and relationship with their children as the most important effect of their children taking medication. This is evident in the following two quotes, where two different mothers highlighted the distress their children and they themselves experienced before starting with the pharmacotherapy and how this was reduced after their children started taking the medication.

*My child’s condition was very severe. She/he had obsessions, she/he could not touch us, we could not touch her/him, we could not hug and kiss her/him, and she/he was obsessed with cleaning. She/he could not get up from her/his seat, she/he could not do any work, and she/he could not eat. She/he could not get out of the bathroom. She/he stayed in the bathroom for 11 hours one day. She/he came out when she/he fainted. She/he would not go to the hospital. I was able to convince her/his to go to the doctor that day. When she/he started taking the medicine, her/his troubles gradually decreased*.(Parent 5)

*She/he seems to have gotten a little better. Previously, my child had attacks in the afternoon. Like during exams. We were going to the emergency room every day. The teachers were calling from school because they had sent [child’s name] to the hospital by ambulance. They [referring to the symptoms] decreased*.(Parent 1)

#### 3.2.2. Theme 2: Disadvantages of Treatment

Although the benefits of medication sounded very promising, the disadvantages and undesirable aspects of treatment were also highlighted. While the subthemes derived from the adolescents’ statements included aspects on the lack of benefit from treatment, possible side effects, a perceived placebo effect of the medication, and feelings of being another person, the subthemes from the parents were on the lack of perceived benefit from treatment and addiction to medication. Therefore, the ‘lack of perceived benefit’ and ‘side effects’ subthemes were presented together. The other subthemes were presented separately, and the subthemes of the adolescents were presented primarily.

##### Lack of Perceived Benefit

Some of the adolescents and parents expressed that they could not perceive the benefits of the medication despite using them regularly, which negatively affected their attitude towards the treatment to the point of not wanting to continue with it anymore. In their opinion, this lack of perceived effect was a major disadvantage of the medication. The adolescents also added that they could not experience a form of ‘healing’, because the medication was ineffective for them. For example, one of the adolescents said that she/he would never get better with the medication, and thus felt like she/he was using it in vain. In addition, the participating adolescents also stated that the medication used did not work because they did not believe in its effect on their bodies and had no hope of being ‘cured’ or ‘healed’ by it. They stated that they believed that their symptoms would not decrease and, therefore, could not feel happy with the medication. According to them, for this reason, they did not see or feel its effect.

*I don’t know. Maybe the medication did not help me much because I do not believe it to be good either. I don’t think that I will get better with medications if I am using them in vain… that my anxieties will pass, that I will be happy. I do not think that I will ever feel happy with the medication*.(Adolescent 3)

Some parents said that there was a benefit of the medication at the start of the pharmacotherapy when the symptoms of their children first started, but that at the present, they could not notice any difference in their children’s condition and symptoms. Similarly, some parents expressed that they did not see the benefits of the medication and were, therefore, hesitant about their children taking it or not. Their children used the medication for a long time, although the expected benefits could not be seen or perceived anymore. One of the parents highlighted that this situation affected the adherence behaviours of the adolescents so that they might stop taking it, regardless of their medical condition.

*I don’t know if the medication is good or bad. It does not seem to help. She/he [referring to the child] has been using the medication for a year, and now we think about stopping it as soon as possible. It is necessary to stop the medication and try to continue without them. I cannot say anything right now because we do not know whether the medicine is helpful or not*.(Parent 9)

##### Side Effects

Most of the adolescents and parents stated that the experienced side effects due to the medication accounted as one of the main drawbacks. The extent of the side effects varied individually. For example, one adolescent described that taking the medication made her/him gain weight, changing her/his physical and body image, which may also have effects on her/his self-esteem. Especially, the school life and work of the adolescents who experienced side effects, such as sleepiness and dizziness, were most affected, as they would fall asleep in classes or miss them completely, because they were so tired.

Some parents described the severity of the side effects their children suffered from the medication. One of the most extreme examples described was a situation where the dosage of the medication had to be changed because the side effects were so severe that they were causing comorbidities. In such cases, parents expressed feelings of ambivalence about the treatment, because while they saw a change in the side effects, the medication was not giving the expected result.

*The doctor had to reduce the dose because the drug caused loss of appetite; that is, it had side effects. Side effects hindered the use of the medication. At that time, the effect of the treatment waned. As a partial solution, the drug dose can be adjusted*.(Parent 7)

##### Placebo Effect (A Subtheme by Adolescents Alone)

Some of the adolescents reported that the medication they used had a placebo effect. They believed in the effects of the medication; therefore, it was effective. When they would not believe that it would be effective for themselves, the medication would not work. They called this situation a placebo effect.

*I do not know if the medication is beneficial, because I believe the medication is beneficial, that is, whether it is a placebo effect or the real effect of the medication*.(Adolescent 12)

Although some of the adolescents described this situation as a placebo effect directly, some of them were not even sure about how to name such an effect. Since they had uncertainty, some adolescents added that they did not understand whether their feelings of well-being were related to the medication or not. As such, they could not distinguish the real effect of the treatment and rather described it as a possible placebo effect. In fact, when they stopped believing that the medication would be helpful, they did not feel the same positive effect as before.

*I think they call it the placebo effect when you think something good will happen after using it; the medication feels good, even if it may not be a therapeutic effect. For example, I was coming home nervous. I was going to take the medication, and I was going to relax, be calm, and that is what usually happens when I think that. I am not thinking that right anymore, and it [referring to the medication] has not been as comforting as it used to be*.(Adolescent 10)

##### Feelings of Being Another Person (A Subtheme by Adolescents Alone)

Some adolescents stated that they believed that their actions or thoughts did not belong to them when they took the medication. They described this situation as a feeling of being ‘another person’ and felt uncomfortable with this. One of adolescents explained that with medication, she/he behaved like a different person from herself/himself. This feeling was quite misleading for herself/himself and for others, as when she/he would take the medication, she/he would be able to concentrate better without attention or concentration problems at school, but she/he did not feel like himself/herself as she/he used to be much more social and unfocused. This was described by the participant as a form of cheating, because she/he used to be not like that.

*Before I took the medication, I was a different person. When I do something, I feel a little different. I was very distracted without taking the medication; I was talking all the time. Now it is as if I am not in the class and as if someone else, someone who is not distracted and hardworking, is taking the class instead of me. It feels like cheating*.(Adolescent 4)

Some adolescents also stated that they could not fully experience their emotions due to the medication and that they experienced fake feelings or fake happiness. They complained that feeling better under the influence of the medication was a false sensation and that the idea that the happiness they felt was created by the medications they took disturbed them heavily. In addition, it was mentioned that sometimes they could not differentiate between their real feelings and the effects of the medication. A participant explained that she/he sometimes could not understand whether the happiness or sadness she/he felt was her/his own or related to the effects of the medication.

*I feel like my emotions are not my own, as if the medication is giving me fake happiness. It is not me who has a feeling. It is as if everything is fake. I’m laughing but it’s fake. It’s not a complete happiness*.(Adolescent 1)

##### Addiction to Medication (A Subtheme by Parents Alone)

Parents stated that they often think that their children will become addicted to the medication and that the medication cannot control the symptoms after a certain period of time of being taken. They highlighted that such need to increase the dosage of the medication is another undesirable aspect of the treatment. They believe that increasing the medication dose is one sign of addiction, rather than understanding that the ineffectiveness of the medication after using it for some time after starting the treatment may be related to the development of tolerance to the medication.

*I wonder if it [referring to taking the medication] started to become a habit or addiction, or did it become less effective because of addiction? At the moment, she/he [referring to the child] is both combative and constantly behaving oppositely. She/he does not do her/his homework now*.(Parent 7)

Some parents said that they preferred psychotherapy instead of pharmacological therapy. In comparison with the medication making their children addicted, they mentioned that psychotherapy has no disadvantages or adverse effects. In addition, they expressed concerns about the use of medication throughout their children’s life as they mentioned that they fear this can lead to addiction. However, they worried that without medication, their children will not be able to achieve anything in the future. These concerns on the part of the parents are clearly exemplified in the following citation.

*The dose of the medication has been increased. It seems to me that she/he [referring to the child] will always be addicted to the medication. If my child does not take it, she/he will not feel good, so will she/he always be addicted to it*?(Parent 3)

#### 3.2.3. Theme 3: Barriers to Medication Adherence

Under this theme, the barriers faced by the adolescents and parents to adhering to the treatment were examined. In the subtheme of ‘individual barriers’, both the adolescents and the parents said that the most important obstacle was forgetting to take the medications. Under the ‘social barriers’ subtheme, the adolescents explained that their parents’ attitude towards using the medication sometimes was a barrier to adhering to the treatment, while parents described ‘stigmatisation’ as a barrier. Most of the adolescents and parents expressed that they needed some more information when the adolescents started to take a new medication under the subtheme ‘lack of information’. The participants stated that if these barriers could be removed, their nonadherence might decrease.

##### Individual Barriers

The first subtheme focussed on the barriers perceived at the individual/personal level. Some adolescents and parents stated that the adolescents often forgot to take their medication or had trouble remembering their medication doses and times. In addition, the adolescents mentioned that when they forgot to take the medication, they felt worse the next day. Their thoughts, feelings, and behaviours were affected by missing the medication. Some adolescents reported that they did not even know why this occurs, despite their parents and friends reminding them to take it. Some even mentioned the measure of setting alarms to remind themselves. According to the adolescents, reminders from their parents and setting an alarm have helped them lower this individual barrier.

However, some of the parents emphasised that the problem was not only forgetting to take the medication, but sometimes they would even forget that they had already taken it and would therefore take it again. They stated that they feared that their children might overdose because of this forgetfulness. They felt a responsibility to remind their children to take their medication and to check on it even if they were not at home or together with the children.

*She/he [referring to the child] went on a trip; I had to remind her/him of the medications when she/he was on the trip. I wondered if she/he took the medication correctly, or if she/he accidentally took two of them. Sometimes she/he does not remember what she/he took*.(Parent 4)

##### Social Barriers

In this subtheme, some social factors were identified as perceived barriers to the medication regimen, such as attitude and stigmatisation of both diagnosing mental disorders and using psychotropic medication. Some adolescents reported that they were frequently exposed to negative attitudes of their parents that affected their decision-making process towards medication use. Their parents emphasised that using the medication was not right for them, they were too small to use the medication, the medication would have many side effects and disadvantages, and they did not want their children to use the medication. One of the adolescents explained that despite the fact that she/he had positive attitudes towards the medication, she/he felt pressured to stop the medication by her/his parents without her/his autonomy or doctor’s advice.

*My parent thinks that the medication is bad for me. She/he presses me to stop, and I am affected because I do not see the effect yet, so my parent is a bit of an obstacle for me right now. I cannot stop alone; I have to stop using medication with the doctor*.(Adolescent 3)

Parents stated that people often behaved differently towards their children and stigmatised them due to the use of the medication. They always asked themselves why their children were different from their peers or had mental problems, and if their children would feel as different, or would feel down, because of their usage of the medication with the following question.

*Why did she/he [referring to the child] use this medication? Why can she/he not focus? Do her/his friends view her/him differently because she/he is taking medications? During this era, all children are using medications, but we went to so many doctors, and I wonder if she/he feels a deficiency because she/he is using medications*.(Parent 11)

Some parents thought that their children might have problems in their professional life in the future because of the stigma towards medication usage. They thought that the psychiatric medication history might be an obstacle to having a job in a governmental institution. They also added that when they talked about the medication usage of their children in their family circle and even with their spouse and friends, they experienced negative reactions due to the medication use and towards their children. Parents were afraid that their children would be stigmatised by their peers. Such fears and worries are clearly expressed in the quote below, where a parent describes the experience of perceived stigmatisation and her/his worries with respect to her/his child’s future due to the stigma they experience from relatives, friends, and institutions.

*If she/he [referring to the child] wanted to be a civil servant or something, it would be bad if the medication showed up on her/his medical record. If she/he does not use medications, her/his record will be clean. If I told a relative that she/he is using this medication, they would reply, ‘What was the need? Is your child crazy?’ I do not listen to them. My spouse thought so too. I explained to her/him that the medicine is necessary, and she/he understood. She/he said to use it then, but that she/he will come to see the doctor next time too*.(Parent 10)

Some parents expressed their negative attitude as they felt sadness and even shame about the fact that their children were diagnosed with a mental disorder and used medications. They mentioned that they could not understand why their children had these problems and were different from other parents’ children. They sometimes felt unsuccessful because they could not solve their children’s problem without treatment. The following quote expresses a wish that the mental health problem and the medication had not happened.

*Every child is small in the eyes of their parents… Being addicted to medication and controlling her/his behaviour through it honestly makes me sad. I wish she/he [referring to the child] could control her/his behaviour without using the medication, but she/he does not, so we use medications*.(Parent 4)

##### Lack of Information

As a final subtheme, the adolescents and their parents expressed lack of information as a barrier to treatment adherence. Most of them stated that when the adolescents started the new medication, they would have needed more information about the treatment. Almost all of them felt that they needed to be informed in detail about the treatment regimen, why they were taking the medication, the way of using the medication, its effect and side effects, the duration of the effect and treatment, and possible changes in daily life. In addition, most of the parents said that they wanted to receive tips on how to approach their children, the situations that may occur in case of overdose, whether adverse effects are permanent, and the change of the dosage.

The adolescents stated that lack of information was a barrier that they often met in relation to mental health settings, and they needed some explanations to remove this barrier. For example, the adolescent stated that when they would be informed about the side effects of the medication, they could manage the process and continue to use the medication. Sometimes, information that the side effect might disappear after a while would make them more comfortable, and they could continue with the medication.

*The doctor had already said that this [referring to the side effect] could happen in the beginning. In other words, the barrier was removed when the doctor informed me in advance, so knowing it, I continued to use the medication*.(Adolescent 8)

#### 3.2.4. Theme 4: Future Dreams

The adolescents and parents expressed their future plans, worries, and dreams throughout the interviews. Both the adolescents and the parents stated that they were worried about their future, if they would not continue or discontinue the treatment. For example, adolescent 8 expressed that she/he might experience problems in interpersonal relationships, such as family problems, social isolation, and anger management problems, if she/he would discontinue the medications in the future.

*I probably will not feel as good as before when I took the medication. I will have big problems with my family again, very likely. Even if I start my own family, I will probably have problems with them too. I might treat my children badly. I may not get along with my wife*.(Adolescent 8)

Some adolescents stated that if they would discontinue or stop the treatment, the symptoms would appear again, and the situation could get worse. Therefore, they would live with the symptoms or need to be hospitalised in the future. Similarly, some parents were worried because if their children would not take their medications in the future, they would start from the beginning and the symptoms of the disorders would reappear. Therefore, they might have hard days, which they had experienced before. In addition, some of the adolescents added the worry that they might not survive because of addiction to a substance or having symptoms and a worsened prognosis for the next decade, which might lead to experiencing an unstable life.

Both adolescents and parents frequently stated their worries about the educational career of the adolescents and that they might lack academic success, and that they would not get admitted to the high school or university they wanted. If they could not graduate from the university, they could not have an occupation or find a job. The treatment would be reflected in their academic and occupational life. Some of the adolescents who would like to work abroad stated that they might be staying in Turkey if they stopped the medication and would be unsuccessful with their dream.

*With poor mental well-being, the discomfort increases exponentially, to the point of having to be hospitalised. I can’t study. I have trouble focusing. I can’t do very well in the university admission exam. I don’t know if I will go to a bad university or work in a factory, stuck in Turkey. That’s how I see it*.(Adolescent 6)

Most of the parents stated that they wanted the adolescents to complete their high school, be accepted at a university, and graduate from university. The adolescents, and especially those with ADHD, stated that if they would use their medications regularly, their academic success might increase, and it would reflect their future. Adolescents and parents frequently reported their aim to have a profession after completing their education. Some of the adolescents stated that they wanted to be mental health professionals and help other people who have similar experiences in early life. Besides, some of them added that if they might get a job, earn, and spend more money, this would make them happy, as expressed in the following statement.

*If I use the medication regularly, my success will increase. If my success increases, I can get into a good high school, a good university, and I can have a good job. Then I will earn a good amount of money, and then can spend my money wherever I want*.(Adolescent 4)

In addition, some of the adolescents stated that they imagined themselves having a driver’s license and that they would like to drive a car. They also dreamed of getting married and hanging up a happy family painting if they use medications regularly in the future. However, some parents with daughters stated that they have worries that their children would marry and would have children impulsively. Their daughters have been diagnosed with conduct disorders, and they might decide without thinking to marry and have children if they do not continue the treatment.

Parents stated that they think that if their children take their medications regularly, they will be healthier and live without any symptoms. They dream that their children could cope with both mental and daily problems, have higher self-confidence, and stand on their own two feet so that their children would be independent. They stated that they imagine that one day the adolescents will not need their parents or anybody else any longer.

## 4. Discussion

In the present study, we examined the perceptions and experiences of not only adolescents with mental disorders but also their parents regarding psychotropic medication in Manisa, Turkey. Our study highlighted the emotional, behavioural, social, and cognitive benefits of psychotropic medication, which drew a finding similar to that of the systematic review by McMillan et al. [8]. However, the results of our study also described the possible disadvantages, barriers to taking or adhering to medication, and future perspectives of the participants themselves. Although these aspects are discussed as distinct themes, some relations between disadvantages of medication use exist with topics that are discussed under barriers to taking or adhering to medication.

While the benefits mentioned focus on symptom reduction and health improvement, the interviews primarily revolved around perceived disadvantages and barriers to medication intake and adherence. One of the main disadvantages mentioned was not being able to perceive or directly see the effect of the medication. This result is consistent with the findings of McMillan et al. [8], which show that the hope for the mental disorder in question to be effectively treated fades when a medication does not work as expected.

In addition, side effects were mentioned as additional disadvantages of treatment and were also described as potential barriers to medication intake and adherence. Although it is well known that psychotropic medications have common side effects, it is often overlooked that young people are more susceptible to them than adults [22,23]. The systematic review by McMillan et al. [8] reaffirms our findings, documenting 18 studies describing experiences with side effects and how, in some cases, this can lead to medication aversion. As a solution to this, the participants in our study mentioned that mental health professionals should follow up and work together with adolescents and their parents and educate them about the specifics of the mental disorder and the medication therapy. This solution could take place in the form of psychoeducational initiatives, such as psychoeducational programmes [4,24].

There were four disadvantages of psychotropic medication that were only mentioned by the adolescent participants and which may therefore be direct expressions of the experience of taking medications, such as describing feelings as ‘false’ or ‘feeling like being someone else’ when taking the medication and what was described as a ‘placebo effect’. Similar descriptions regarding the first one have been found in the literature, with participants taking psychotropic medication describing themselves or their feelings as ‘weirdness’ [25], ‘acting differently’ [26], or ‘not presenting their authentic self’ [27]. Furthermore, what was described in our study as a placebo effect relates to the adolescents being unsure whether the medication actually had an effect or it was their own attitude towards it that made them believe it was working. This placebo effect was also pointed out by McKinney and Greenfield in their study, in which some of the participants who started to take psychotropic medication began to feel much better even before the expected therapeutic effect period, which the authors described as a placebo effect [28].

Barriers to medication adherence included forgetting to take the medication or the medication dosage. Nevertheless, receiving reminders from parents was perceived as another barrier towards medication intake. Here again, McMillan et al. [8] highlighted how medication reminders from parents can be compelling for adolescents and can undermine their decision-making capacity and autonomy. Solutions provided directly by adolescents vary in cognitive and behavioural techniques, such as setting alarms or reminder notes that help them not only to take their medication but also to keep the proper medication dosage [24,29].

The disadvantages mentioned so far focus on the unwanted effects of medication. However, it could be argued that at the root of these perceptions and experiences is a lack of information and low levels of knowledge and/or awareness of the issue, as this may influence what is or is not expected from psychotropic therapy. For example, the fear of developing an addiction to medication was highlighted as another disadvantage. Especially parents thought that medication led to addiction, because the dosage was always changed, the medication was not as effective as at the beginning of therapy, or they perceived that it did not work at all. Together with the other disadvantages described, this perceived disadvantage might have its origin in the lack of knowledge about mental disorders and pharmacotherapy [3]. In this sense, psychoeducational initiatives could be essential interventions for adolescents and their families. These could address psychotropic medication in general, ways of taking it, the possible development of tolerance, and the therapeutic and adverse effects of a treatment plan, as also suggested by other literature [3,24]. It is relevant to note here that one dimension of the World Health Organisation’s adherence model [11] emphasises the importance of educational training for long-term adherence to therapies, as well as patient follow-up. However, it is not only the individual that needs to be considered, but also the health system as a whole, so the importance of structural issues, such as availability of and access to and capacity of (mental) health services, also needs to be emphasised. In practice, this is often difficult to implement: in Turkey, for example, there is no regular follow-up, and the time spent with each patient is only about 5–6 min. Therefore, (mental) health professionals do not find enough time to train their patients. Considerations for improvement therefore should not remain at the individual level, but include policy and structural changes that promote and improve mental health and mental health services for children and adolescents.

On a more social level, negative attitudes and stigmatisation have been shown as social barriers to medication intake and adherence [8,25]. In our study, while adolescents mentioned their parents’ negative attitude towards treatment, the parents reported stigmatisation and discrimination of their children in their social environment. The attitude of parents or the family towards mental disorders and/or medication or therapy has been found to determine the adherence of adolescents [8,30]. Given that only parents reported some experiences of stigmatisation, it is possible that they were more sensitised to or directly confronted with it. When it was the parents themselves who displayed negative attitudes or internalised stigma at a personal level about medication or mental disorders, it is possible that these perceptions or attitudes may have developed to protect their children and family from stigma in their social circle [3,8]. We assume that feelings such as sadness and shame for the diagnosis of mental disorders or wishes that their children do not have to rely on medication might reflect a nonacceptance of their children’s mental disorder and treatment. Similarly, Heary et al. [31] emphasised that children and adolescents with mental disorders could be exposed to a variety of implicit messages in their family environment. Adolescents could be influenced by these messages and could internalise the beliefs of their parents. Structured programmes should thus focus on not only adolescents with mental disorders but also their parents and/or families in order to reduce stigmatisation. One of the other suggested strategies would be the contact-based model, where information is delivered by someone who has experienced mental illness, and which has been found to be effective among adults [32]. There is a strong need for interventions to be tailored to the developmental stage of adolescents [31]. Such psychoeducational efforts could also target the general population through information campaigns on mental disorders as a first step to raise public awareness on the topic and decrease social stigmatisation and internalised stigmatisation.

While, so far, the direct effects of perceived barriers to medication intake and adherence have been discussed, our study also looked at concerns regarding long-term consequences of mental disorders and their treatment with medication. Both the adolescents themselves and their parents expressed concerns about future prospects, including aspects such as further education and vocational training. Again, these results are consistent with the systematic review by McMillan et al. [8]. The authors described that the studies included in their review referred to how young people who used psychotropic medication and experienced health improvements from it looked forward to the future and their plans [8]. Specific to our study, it would be essential to highlight to what extent the expressed future prospects were in line with traditional socio-culturally and religiously shaped expectations about young adults in Turkey, such as getting married after graduation and finding a job. Regarding marriage prospects, only the parents of the daughters mentioned their concern about marriage. It can be assumed that this gender difference could be based on the fear of precocious marriage among girls, as is the social reality in the Middle East [33]. Additionally, the adolescent girls themselves were concerned about future issues related to interpersonal relationships in their marriage and family. These results again highlight the importance of sociocultural, gender, and religious aspects and how they shape social expectations [34].

In addition, the future perspectives of parents for their children included components of independence and not being dependent on parental support. A study by Saritas and Dikec [35] describes similar concerns, in which parents of children with mental disorders and/or disabilities worried about what would happen if they were to die before their children, as their children would be left without supportive assistance. Especially in contexts where deficits in health care and social services for people with mental disorders and disabilities exist, as in Turkey, family support is of utmost importance. We recommend further research, but also public health action in this direction, as the current socioeconomic and pandemic crises, not only in Turkey but worldwide, could lead to a worsening of the situation. It is also indicated by this unexpected finding of our study that one-half of the adolescent participants reported a suicide attempt, and this high rate could be interpreted as an effect of this crises.

## 5. Strengths and Limitations of the Study

The present study paves the way for future research by allowing the participation not only of the adolescents but also of their parents. The triangulation of the experiences and perceptions of both adolescents and their parents allowed for a more comprehensive perspective on the topic. Following Colaizzi’s phenomenological method of interpretation, it was possible to include both groups in a participatory way. However, some limitations of this study must also be taken into account. First, the results of this study cannot be generalised, as they are limited to the experiences and perceptions of 24 participants. The same applies to the experiences and perceptions of fathers, as they may be under-represented, given that most of the parents included were mothers. For future studies, we would like to encourage a higher participation of males. In addition to these limitations, this study used a purposive sampling method, which is very prone to researcher bias, because the sample created depends on the judgement of the researcher. However, as we used widely accepted criteria for sample selection, this bias is likely to be limited. Furthermore, as in any other cross-linguistic studies, there may be translation problems with the original Turkish data, as meanings may be lost or changed in the translation of quotes or statements. Lastly, the fact that the interviews were conducted by a trained psychiatrist might have introduced a preconception of a positive connotation of adherence to psychotropic medication. This needs to be taken into account when interpreting the data.

## 6. Conclusions

In order to develop effective and useful measures to improve adherence to psychotropic medication and reduce health expenditure among adolescents, it is necessary to understand what influences medication adherence or nonadherence among adolescents with mental disorders and their families. In this sense, our study highlights the main positive effects of psychotropic medication and barriers to medication intake and adherence. The positive effects included symptom management and health improvement. The barriers varied from those directly linked to the medication effects (e.g., negative side effects, lack of perceived effect) to personal barriers (e.g., forgetting to take medication or feelings of not being oneself due to medication intake) and societal ones. Regarding the latter, it was mentioned that sociocultural and religious factors, as well as gender-related factors, strongly influence perceptions of mental disorders and pharmacotherapy. This was reflected in long-term concerns and future perspectives along the lines of taking medication, as well as medication dependence.

Recommendations from the side of the participants included evidence-based psychoeducation programmes, where a mental health professional would work together with adolescents and parents to inform and create awareness about mental disorders and treatment options. This solution could well be adapted to a more societal level, where psychoeducational campaigns could fight social stigmatisation. Initiatives, such as the recommendations proposed, could demonstrate the first steps towards a more inclusive care of mental disorders. However, changes on such small scales cannot reach their full impact without the support of social and political structures that fully acknowledge the needs and rights of people with mental disorders and/or disabilities and their caregivers.

## Figures and Tables

**Table 1 ijerph-19-09589-t001:** Characteristics of the participants.

	Adolescents		Parents	
	Mean	SD	Mean	SD
**Age**	14.17	1.46	42.42	6.09
**Gender**	**N**	**%**	**N**	**%**
Female	9	75	10	83.3
Male	3	25	2	16.7
	**Attending School**		**Employment**	
Yes	12	100	6	50
No			6	50
**Economic Status**				
Low	1	8.3	1	8.3
Moderate	8	66.7	10	83.4
High	3	25	1	8.3
**Smoking**				
Yes	2	16.7	5	41.7
No	10	83.3	7	58.3
**Alcohol Use**				
Yes	1	91.7	1	8.3
No	11	8.3	11	91.7
**Substances Use**				
No	12	100	12	100
**Physical Illness**				
Yes	3	25	7	58.3
No	9	75	5	41.7

SD: standard deviation.

**Table 2 ijerph-19-09589-t002:** Mental health history of the adolescents.

Suicide Attempt	*n*	%		
Yes	6	50		
No	6	50		
**Hospitalisation**				
Yes	1	8.3		
No	11	91.7		
**Treatments**				
Antipsychotics + antidepressants	2	16.7		
Antidepressants	7	58.3		
Psychostimulants	3	25		
Psychostimulants + antipsychotics	1	8.3		
**Mental Disorders**				
Attention deficit hyperactivity disorder	3	25		
Depressive disorder	4	33.4		
Anxiety disorder	4	33.4		
Obsessive–compulsive disorder	1	8.3		
	**Mean**	**SD**	**Min**	**Max**
**Duration of Mental Disorders (Months)**	24.0	27.78	2	84
**Duration of Treatment (Months)**	14.33	15.38	3	48

SD: standard deviation; Min: minimum, Max: maximum.

**Table 3 ijerph-19-09589-t003:** Themes of the adolescents and their parents.

Adolescents	Parents
**Theme 1: Benefits of Treatment**	
**Theme 2: Disadvantages of Treatment**	
Lack of Perceived Benefit	
Side Effects	
Placebo Effect	Addiction on Medication
Feeling of Being Another Person	
**Theme 3: Barriers to Medication Adherence**	
Individuals Barriers	
Social Barriers	
Lack of Information	
**Theme 4: Future Dreams**	

## Data Availability

The data presented in this study are available on request from the corresponding author.

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
