# Peer review of "Perceptions and Experiences of Adolescents with Mental Disorders and Their Parents about Psychotropic Medications in Turkey: A Qualitative Study"

_ijerph, 2022, doi:10.3390/ijerph19159589_

Round 1

Reviewer 1 Report

To the authors:

Thank you for the opportunity to read and review the article titled Perceptions and Experiences of Adolescents with Mental Dis-orders and their Parents about Psychotropic Medications in Turkey: A Qualitative Study for the IJERPH. I find the article to be within the scope of IJERPH, and the topic should be of interest to the journals’ readers.

Overall, I really enjoyed reading your manuscript, the topic is important and relevant, it is well written, and with acceptable quality with some exceptions. Most importantly, you should elaborate on research ethics, including general data protection management and confidentiality measures. You have collected personal data amongst adolescents, which may introduce additional rights and additional measures for safeguarding? Furthermore, I have a few questions, and would like to suggest some minor revisions of the manuscript.

Introduction:

·         In line 46-55, factors for non-adherence are described; 1) socio-demographic/cultural, 2) clinical, 3) medication-related, and 4) patient- or family related. According to WHO, there are five dimensions affecting adherence. Is it feasible also to describe, in brief, the fifth dimension; healthcare team/system related factors, relevant for mental illness amongst adolescents? I think this would be of interest, also as you highlight the need for research in the Non-Western/Turkey context. Are there system factors in Turkey that is relevant for psychotropics adherence amongst adolescents?

·         In line 70-71, the systematic review by McMillan et al., 2022, is mentioned as “one of the few publications that focus on medication adherence among adolescents with mental disorders”. I find that there are many articles on this topic published, and you refer to other reviews that proves that (e.g., Edgcomb & Zima 2018, Häge et al., 2018). I suggest rewriting this part. It might be suitable to emphasize that McMillan et al., 2022, is one of the few systematic reviews that focus on qualitative evidence on medication adherence among adolescents with mental disorders?

Method:

·         I suggest that you include a section on ethics in the Method chapter. Please describe the personal data protection management, confidentiality measures, and overall ethical issues relevant, and how they were managed.

·         COREQ was used in the reporting of this study. I suggest that the checklist is filled out and reported as an appendix. Can you make sure to include all relevant information, since I cannot find information on the following COREQ items:

o   2) researcher’s credentials,

o   7) what did the adolescents/parents know about the researcher?

o   8) characteristics about the interviewer (bias, assumptions, reasons, interests? Why did you use a psychiatrist?)

o   11) how were participants approached?

o   13) refused to participate/drop-outs?

o   15) was anyone else present besides the participant and researcher/psychiatrist (this is very relevant when interviewing children/adolescents)?

o   17) did you pilot the interview guide?

o   20) Field notes?

o   22) Discussion regarding data saturation?

·         A semi-structured interview guide was developed according to the literature. Could you describe the development process (who were involved, pilot, and so on?) and literature used? How is reference 4 and 5 relevant?

·         The interview guide: could you give examples of questions used in the fourth category (perceptions, opinions, and experiences with the use of psychotropic medications)? Or consider including the interview guide as an appendix.

·         How was data saturation discovered and analysed?

Results:

·         Regarding the participants characteristics: Do you have information about parents’ mental health (since you have info on physical illness)? Do you have info on medication use of parents? I believe this is relevant for their beliefs about medicines.

·         Table 1: How was “substances use” defined?

·         Table 2: The minimum duration of treatment is given at 1 month. However, one inclusion criterion for study participation was the use of psychotropic medications for at least three months (line 118-119). Is the result correct? Or the inclusion criterion incorrect? Or do you need to exclude one of the adolescent and parent?

·         You may consider introducing a quotation so support the results of all subthemes (for example missing in Side Effects), or present a coding tree.

·         I suggest that you change the quotation identifier of the parents: use Parent 1 to 12 instead of mother/father. Since only two fathers are participants, their confidentiality may be at risk?

·         Sometimes the results are presented as he/she and him/herself, for example in the subtheme “Feelings of Being Another Person”. Is it deliberate to hide the participants gender? It should be consistent throughout the Result chapter?

·         Would it be feasible to elaborate on the relationship between the main themes? For example, there are possible relationships between Theme 2 and 3, as all subthemes of theme 2 may also be considered as barriers to medication adherence? For example, I find some of the results presented in disadvantages of treatment to also be about adherence barriers (e.g., Line 254: “not wanting to continue”, Line 272: “hesitant about their children taking it or not”). In the subtheme “Individual barriers”, results regarding mostly unintentional adherence is presented (forgetfulness). However, expectations of several barriers are provided in the result material, for example side effects. An explanation of the main theme relationships may increase the trustworthiness of your results.

Discussion:

·         The first section of the discussion (line 519-524) can be deleted? If included, it should be discussed regarding your results.

·         I find beliefs about medicines to be a very central aspect of your discussion, and that adolescents and their parents’ beliefs are both coincident and somewhat different. However, you do not elaborate on different strategies, rather mention psychoeducation initiatives in general for adolescents and their families. Would it be of interest to discuss strategies tailored to the separate populations? And discuss behavior as well, since unintentional adherence barriers were highlighted?

·         Line 549-552: two disadvantages of psychotropic medicines that were only mentioned by adolescents are discussed, the first being forgetting to take the medication. Is this correct? According to Table 3, these subthemes were “Placebo effect” and “Feeling of being another person”? The discussion should probably be revised in this part.

·         Limitations of the study: I would like to read your thoughts on how using a psychiatrist and not the researchers as interviewer, could affect the results. What are the limitations of your sampling strategy?  

Author Response

Thank you for the opportunity to read and review the article titled Perceptions and Experiences of Adolescents with Mental Disorders and their Parents about Psychotropic Medications in Turkey: A Qualitative Study for the IJERPH. I find the article to be within the scope of IJERPH, and the topic should be of interest to the journals’ readers. Overall, I really enjoyed reading your manuscript, the topic is important and relevant, it is well written, and with acceptable quality with some exceptions.

Thank you for your contributions.

Most importantly, you should elaborate on research ethics, including general data protection management and confidentiality measures. You have collected personal data amongst adolescents, which may introduce additional rights and additional measures for safeguarding?

We have added ethical issues at the method and add more details on this part (line 192-199).  

Furthermore, I have a few questions, and would like to suggest some minor revisions of the manuscript. 

Introduction:

In line 46-55, factors for non-adherence are described; 1) socio-demographic/cultural, 2) clinical, 3) medication-related, and 4) patient- or family related. According to WHO, there are five dimensions affecting adherence. Is it feasible also to describe, in brief, the fifth dimension; healthcare team/system related factors, relevant for mental illness amongst adolescents? I think this would be of interest, also as you highlight the need for research in the Non-Western/Turkey context. Are there system factors in Turkey that is relevant for psychotropics adherence amongst adolescents?

We added the missing dimension to the introduction (line 49-52) and further discussed this dimension with our data in the first paragraph of discussion on page 15 (line 562-565).

In line 70-71, the systematic review by McMillan et al., 2022, is mentioned as “one of the few publications that focus on medication adherence among adolescents with mental disorders”. I find that there are many articles on this topic published, and you refer to other reviews that proves that (e.g., Edgcomb & Zima 2018, Häge et al., 2018). I suggest rewriting this part. It might be suitable to emphasize that McMillan et al., 2022, is one of the few systematic reviews that focus on qualitative evidence on medication adherence among adolescents with mental disorders?

We revised this sentence according to your suggestion; “One of the few systematic reviews that focuses on qualitative evidence on medication adherence among adolescents with mental disorder is a study by McMillan et al.” (line 71-72).

Method:

·        I suggest that you include a section on ethics in the Method chapter. Please describe the personal data protection management, confidentiality measures, and overall ethical issues relevant, and how they were managed.

COREQ was used in the reporting of this study. I suggest that the checklist is filled out and reported as an appendix. Can you make sure to include all relevant information, since I cannot find information on the following COREQ items:

We added an “ethical consideration” part in the methods section and gave more details about personal data protection, ethical and institutional permissions and other ethical issues.

We filled the COREQ checklist and attached it as an appendix.

researcher’s credentials,

The researchers credentials were added in the data collection part (lines 153-157). We explained why we used a psychiatrist as an interviewer: A team member not known to the interviewees who had knowledge about qualitative studies was selected so that adolescents and their families could openly express their feelings and thoughts. The interviewer, is trained in therapeutic interviews with children and adolescents with mental disorders and their families in a university hospital in Turkey and conducted all in-depth interviews without another person being present.”

what did the adolescents/parents know about the researcher?

characteristics about the interviewer (bias, assumptions, reasons, interests? Why did you use a psychiatrist?)

how were participants approached?

refused to participate/drop-outs?

We added this explanation at the beginning of the results section: One adolescent and one parent dropped out of the study. While they initially accepted to participate in this study in the first session, they did not attend the second one. Also, one parent was interviewed and her/his child refused to participate.” (lines 202-204)

was anyone else present besides the participant and researcher/psychiatrist (this is very relevant when interviewing children/adolescents)?

We have included the description in line 158-160.

did you pilot the interview guide?

Yes, we did a pilot interview with two adolescents and their parents. We added this information in the data collection part (line 146-148): After developing the interview guide, pilot interviews were conducted with two adolescents and their parents. These were not included in the data of our study. No changes to the interview guide were done after the pilot interviews.”

Field notes?

We added this information at the end of the data collection part (line 168-170): When the data started to become repetitive, in other words, when the data reached saturation, no more interviews were conducted. Field notes were not made during the interviews.”

Discussion regarding data saturation?

How was data saturation discovered and analysed?

A semi-structured interview guide was developed according to the literature. Could you describe the development process (who were involved, pilot, and so on?) and literature used? How is reference 4 and 5 relevant?

We have explained the interview guide  and the relevance of references 4 and 5 as well as some example questions of the last category in more detail in the methods section (lines 140-145).

The interview guide: could you give examples of questions used in the fourth category (perceptions, opinions, and experiences with the use of psychotropic medications)? Or consider including the interview guide as an appendix.

Results:

·         Regarding the participants characteristics: Do you have information about parents’ mental health (since you have info on physical illness)? Do you have info on medication use of parents? I believe this is relevant for their beliefs about medicines.

We mention the inclusion criterion for the parents as follows (lines 122-123): not diagnosed with a mental disorder according to the DSM-V were eligible to participate“. Therefore, parents with mental disorders and/or those taking psychotropic medications were excluded in this study.

Table 1: How was “substances use” defined?

We asked the adolescents directly about alcohol and substance use, no measurement was used to collect information on substance use. We added this clarification as follows (lines 138-140):In the third category, the participants were asked directly about substance and alcohol use. No measurement scale was used to collect substance use information.”

Table 2: The minimum duration of treatment is given at 1 month. However, one inclusion criterion for study participation was the use of psychotropic medications for at least three months (line 118-119). Is the result correct? Or the inclusion criterion incorrect? Or do you need to exclude one of the adolescent and parent?

Thank you for your attention, the value in Table 2 was incorrect and we revised it. The inclusion criterion was correct. 

 You may consider introducing a quotation so support the results of all subthemes (for example missing in Side Effects), or present a coding tree.

As there is no rule requiring an example quote for each theme and subtheme, we avoided to repeat the same issues when writing the results instead. Thank you for your suggestion, we have now added a quote to support this subtheme (lines 336-338).

I suggest that you change the quotation identifier of the parents: use Parent 1 to 12 instead of mother/father. Since only two fathers are participants, their confidentiality may be at risk?

Thank you for your suggestion, we wrote now parent instead of mother or father.

Sometimes the results are presented as he/she and him/herself, for example in the subtheme “Feelings of Being Another Person”. Is it deliberate to hide the participants gender? It should be consistent throughout the Result chapter?

Thank you for your attention, we rewrote direct gender pronouns as: “she/he”, “her/his”, “herself/himself” throughout the whole manuscript.

Would it be feasible to elaborate on the relationship between the main themes? For example, there are possible relationships between Theme 2 and 3, as all subthemes of theme 2 may also be considered as barriers to medication adherence? For example, I find some of the results presented in disadvantages of treatment to also be about adherence barriers (e.g., Line 254: “not wanting to continue”, Line 272: “hesitant about their children taking it or not”). In the subtheme “Individual barriers”, results regarding mostly unintentional adherence is presented (forgetfulness). However, expectations of several barriers are provided in the result material, for example side effects. An explanation of the main theme relationships may increase the trustworthiness of your results.

Thank you for pointing such inter-relationships between the main themes out. We have therefore included a statement at the beginning of the discussion on such possible inter-relationships (lines 562-564): “Although these aspects are discussed as distinct themes some relations between disadvantages of medication use exist with topics that are discussed under barriers to taking or adhering to medication.”

Discussion:

·         The first section of the discussion (line 519-524) can be deleted? If included, it should be discussed regarding your results. 

The first paragraph was deleted, and we started the discussion directly with our results.  

I find beliefs about medicines to be a very central aspect of your discussion, and that adolescents and their parents’ beliefs are both coincident and somewhat different. However, you do not elaborate on different strategies, rather mention psychoeducation initiatives in general for adolescents and their families. Would it be of interest to discuss strategies tailored to the separate populations? And discuss behavior as well, since unintentional adherence barriers were highlighted?

We discussed the attitudes and beliefs of parents and adolescents in the discussion part on social barriers issues (lines 641-644) and added new references (lines 641-652) to offer other interventions to reduce stigmatization among adolescents.  

Line 549-552: two disadvantages of psychotropic medicines that were only mentioned by adolescents are discussed, the first being forgetting to take the medication. Is this correct? According to Table 3, these subthemes were “Placebo effect” and “Feeling of being another person”? The discussion should probably be revised in this part. 

Thank you for your attention. We revised this part accordingly in the results (lines 582-585).

Limitations of the study: I would like to read your thoughts on how using a psychiatrist and not the researchers as interviewer, could affect the results.

What are the limitations of your sampling strategy?  

Thank you for this comment. We have now added a statement to the limitation section (lines 699-701): “Lastly, the fact that the interviews were conducted by a trained psychiatrist might have introduced a preconception of a positive connotation of adherence to psychotropic medication. This needs to be taken into account when interpreting the data.”

Thank you for this comment. We have added this sentences (line 693-696); In addition to these limitations, this study used a purposive sampling method, which is very prone to researcher bias, because the sample created depends on the judgement of the researcher. However, as we used widely accepted criteria for sample selection, this bias is likely to be limited.”

Reviewer 2 Report

Congratulations on the research. Studying the experiences of patients with mental health illnesses is important to improve their treatment and optimize resources.

Here are some recommendations:
- Describe who the interviewers were and whether they had had prior training to avoid bias.

- In the table 1,  describe which diseases are included in the Physical Illness variable.

Author Response

Congratulations on the research. Studying the experiences of patients with mental health illnesses is important to improve their treatment and optimize resources.

Thank you for your contributions.

Here are some recommendations:
- Describe who the interviewers were and whether they had had prior training to avoid bias.

It were added in the data collection part (lines 153-157). We explained why we used a psychiatrist as an interviewer: A team member not known to the interviewees who had knowledge about qualitative studies was selected so that adolescents and their families could openly express their feelings and thoughts. The interviewer, is trained in therapeutic interviews with children and adolescents with mental disorders and their families in a university hospital in Turkey and conducted all in-depth interviews without another person being present.”

- In the table 1, describe which diseases are included in the Physical Illness variable.

We have written the physical illness diagnoses in the results section under the participants’ characteristics (line 218-129 and 225-226).